# ATTENTION CALIBRATION FOR REDUCING HALLUCI-NATION IN LARGE VISION-LANGUAGE MODELS

## ABSTRACT

Large Vision-Language Models (LVLMs) exhibit impressive multimodal reasoning capabilities but remain highly susceptible to object hallucination, where models generate responses that are not factually aligned with the visual content. Recent works attribute this issue to an inherent bias of LVLMs where vision token attention map has spurious focus on certain positions, and propose to mitigate this issue by reordering visual tokens. However, we find that different LVLMs exhibit different correlations between attention and spatial position, which makes the existing static solution difficult to generalize to other LVLMs. To begin with, we investigate the attention bias introduced by image tokens through a toy experiment, in which a blank image is fed into the model to capture its position-dependent bias. We then remove this bias from the original attention map, which already leads to a substantial reduction in hallucinations. This proof of concept validates the core intuition behind attention calibration. Building upon this insight, we propose Dynamic Attention Calibration (DAC)—a lightweight, plug-and-play module that leverages contrastive learning to dynamically enforce positional invariance. Unlike static baselines, DAC adapts to different models and inputs in a robust and learnable manner, offering a generalizable solution to mitigate attention-related hallucinations in LVLMs. Comprehensive experiments across multiple benchmarks demonstrate that DAC significantly reduces object hallucination while improving general multimodal alignment. Our method achieves state-of-the-art performance across diverse LVLM architectures on various metrics.

## 1 INTRODUCTION

Large Vision-Language Models (LVLMs) Liu et al. (2024d); Bai et al. (2023); Dai et al. (2024); Zhu et al. (2023); Ye et al. (2024) have garnered significant attention in the AI research community for their remarkable ability to comprehend the visual world and engage in conversational interactions with humans. Despite these advances, LVLMs continue to face critical challenges, particularly in the form of object hallucination Li et al. (2023b); Rohrbach et al. (2018); Cui et al. (2023), a phenomenon where models generate responses that are not factually aligned with the visual content. This issue undermines the reliability of LVLMs, posing a significant barrier to their deployment in real-world applications.

A variety of approaches have been proposed to mitigate object hallucination in LVLMs. One common strategy involves post-hoc correction using revisor models Yin et al. (2023); Zhou et al. (2024); Lee et al. (2023), which aim to reduce hallucinated responses by refining outputs. Another approach improves supervised fine-tuning through diversified instruction tuning data Liu et al. (2024a); Yu et al. (2024) or aligns model responses with human preferences Sun et al. (2023). Recently, several studies have explored training-free methods for mitigating object hallucination by addressing issues in the autoregressive decoding process of LVLMs Leng et al. (2023); Huo et al. (2024); Huang et al. (2023).

A recent study Xing et al. (2024) reveals that LVLMs' perception varies with object positions due to the inherent processing order in autoregressive models. As 2D vision tokens are concatenated with text tokens and flattened into a raster-scan sequence (top-to-bottom, left-to-right), the model develops a bias, prioritizing tokens in the bottom-right region closer to the instruction tokens (Figure 1a), termed as Spatial Perception Bias (SPB). This spatial bias skews perception capabilities. To mitigate

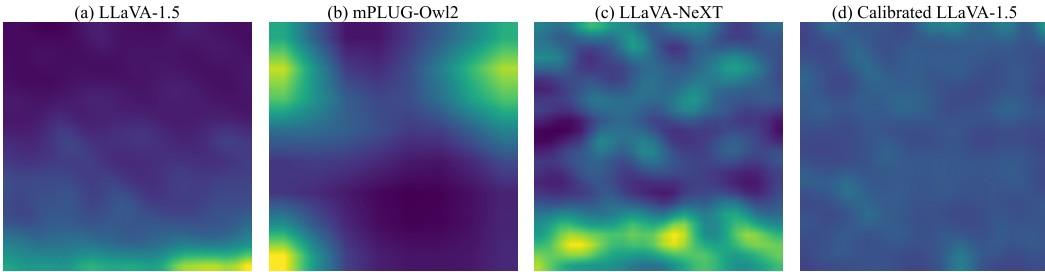

Figure 1: Spatial Position Bias influences how LVLMs perceive objects based on their position within an image. The visualization above illustrates vision token attention weights from the final token before output generation, during the decoding process for different models on a blank white image in response to the open-ended prompt: "Please describe this image in detail." (a) shows LLaVA-1.5, which exhibits an increasing trend in attention distribution following a raster scan order, as identified by Xing et al. (2024). (b-c) represent other models, displaying arbitrary attention distributions. (d) depicts the calibrated vision token attention map of LLaVA-1.5 after Dynamic Attention Calibration.

this, Xing et al. (2024) propose a position alignment technique that reorders the perception sequence, reducing spatial bias.

However, this approach has two major limitations. First, the method is based on the assumption that the model assigns greater attention to tokens that are relatively nearby. As demonstrated in Figure 1(a-c), our analysis reveals that the attention distributions of vision tokens vary significantly across different LVLM models and unexpectedly high attentions are assigned to arbitrary locations. This observation challenges the generalization of the heuristic reordering strategy proposed by Xing et al. (2024), highlighting the need for a more dynamic and adaptable solution. Second, the proposed technique requires retraining the entire network, which is computationally expensive and often impractical for large-scale LVLMs, underscoring the necessity of developing a lightweight alternative.

Building on this analysis, we propose to mitigate object hallucination by calibrating the SPB in attention maps. As a proof of concept, Uniform Attention Calibration (UAC) subtracts a static bias extracted from the attention map of a blank image input and confirms that reducing SPB lowers hallucination. Motivated by this evidence, we further relax the assumption in UAC and introduce Dynamic Attention Calibration (DAC) to fine-tune LVLMs for better generalization. Specifically, DAC consists of a learnable plug-and-play module integrated into the self-attention mechanism. With a simple yet effective data augmentation technique, the module is then fine-tuned via contrastive learning to encourage consistent outputs with different object positions in the image, which dynamically adjusts vision token attention map to tackle object hallucination.

Comprehensive experiments confirm the effectiveness of DAC, revealing substantial improvements across multiple object hallucination benchmarks for a range of LVLMs, including LLaVA-1.5 Liu et al. (2024d), mPLUG-Owl2 Ye et al. (2024), and LLaVA-NeXT Liu et al. (2024c). Additionally, our approach strength- ens the overall perception capabilities of LVLMs, as demonstrated by strong performance on MME Fu et al. (2024) and LLaVA-Bench Liu et al. (2024d). In summary, our main contributions are as follows:

1. We systematically investigate Spatial Perception Bias (SPB) in the attention mechanism of various LVLMs, revealing its strong correlation with object hallucination and its unpredictable nature across different models.

2. We propose Dynamic Attention Calibration (DAC), a lightweight, learnable, and plug-and-play module that dynamically adjusts vision token attention to robustly mitigate SPB.

3. Extensive experiments confirm that DAC significantly reduces object hallucination and enhances overall perception, achieving notable improvements across multiple LVLMs and benchmarks.

## 2 RELATED WORK

### 2.1 VISUAL-LANGUAGE MODELS

Large Vision-Language Models (LVLMs) have evolved from early BERT-based architectures Devlin et al. (2018); Lu et al. (2019); Chen et al. (2019) to models that integrate Large Language Models (LLMs) Bai et al. (2023); Brown et al. (2020); Gilardi et al. (2023); Raffel et al. (2020); Taori et al. (2023). Early vision-language models, such as ViLBERT Lu et al. (2019) and LXMERT Tan & Bansal (2019), fused visual and textual features through transformer-based architectures. The introduction of LLMs enabled contrastive learning approaches like CLIP Radford et al. (2021) and ALIGN Jia et al. (2021), improving multimodal adaptability. Recent LVLMs, such as LLaVA Liu et al. (2024d) and InstructBLIP Dai et al. (2024), leverage visual instruction tuning for improved context-aware generation. Advances have further enabled referential dialogues Chen et al. (2023a); You et al. (2023); Zhang et al. (2023a), interleaved image-text processing Alayrac et al. (2022); Awadalla et al. (2023), and visual prompts Peng et al. (2023); Zhang et al. (2023b); Chen et al. (2023b), broadening LVLM applications in interactive AI systems. These developments highlight a growing shift toward task-specific fine-tuning and multimodal interaction.

### 2.2 HALLUCINATION IN VLMS

Object hallucination arises when Large Vision-Language Models (LVLMs) generate textual descriptions containing objects or attributes not present in the accompanying image Cui et al. (2023); Liu et al. (2024b); Guan et al. (2023); Li et al. (2023a); Wang et al. (2024); Nie et al. (2024). This phenomenon is frequently observed in tasks such as image captioning and visual question answering, where maintaining an accurate alignment between visual and textual content is critical. A range of methods has been proposed to address hallucination, from post-hoc correction using external or self-correcting models Yin et al. (2023); Zhou et al. (2024); Lee et al. (2023) to enhanced instruction tuning that diversifies training data or aligns outputs with human feedback Liu et al. (2024a); Yu et al. (2024); Sun et al. (2023). Recently, training-free approaches that rely on model-based distribution comparisons were proposed Leng et al. (2023); Huo et al. (2024); Huang et al. (2023). As LVLMs grow more sophisticated and versatile, understanding and mitigating object hallucination remains a key focus in multimodal learning research. From a unique perspective, our design is rooted in the correlation between vision token attention and object hallucination.

## 3 PRELIMINARY

In this section, we provide a brief overview of the widely adopted LVLMs architecture and explain how vision tokens are involved in the self attention module. Additionally, we review the how LVLMs exhibit spatial perception bias problem, highlighting systematic biases that affect LVLM hallucination.

### 3.1 LVLMS: GENERATION AND ATTENTION MECHANISM

**Vision and Language Inputs** LVLMs process both image $v$ and text $t$ inputs. Raw images are divided into patches and encoded by a visual encoder, followed by a cross-modal projection module that maps visual features into the token space. This yields a sequence of vision tokens $v = \{v_i \mid i = 1, 2, \ldots, n\}$, where $n$ is the number of vision tokens. Text inputs are tokenized and embedded into text tokens $t = \{t_j \mid j = 1, 2, \ldots, m\}$, where $m$ is the number of text tokens. The vision and text tokens are then concatenated into a unified input sequence $x = \{v, t\}$[1], ensuring a shared multimodal representation space, with $v_i, t_j \in \mathbb{R}^d$, where $d$ denotes the feature dimensionality.

**Language Model Generation** LVLMs are typically built on pre-trained LLMs such as Vicuna Chiang et al. (2023) or LLaMA Touvron et al. (2023), parameterized by $\theta$. The model takes a input $x$ and predicts the next token probability $p(y_i)$ at time step $i$ in an autoregressive manner:

$$p(y_i \mid x, y_{<i}) = \text{softmax}(\text{logit}_\theta(y_i \mid x, y_{<i})) \tag{1}$$

---

[1]We omit the system tokens for simplicity.

**Self-Attention Mechanism** The self-attention mechanism computes token relevance by projecting the output of previous layer into query $Q$, key $K$, and value $V$ with linear transformations $W_Q$, $W_K$, $W_V$. The self attention output is computed as

$$\text{SA}(Q, K, V) = \text{softmax}(\mathbf{A} + M) \cdot V, \ \mathbf{A} = \frac{Q \cdot K^T}{\sqrt{d_l}}, \tag{2}$$

where $\mathbf{A} \in \mathbb{R}^{B \times H \times (n+m) \times (n+m)}$ denotes attention weight matrix, $B$ and $H$ represent the batch size, and number of attention heads, respectively. $M$ denotes the causal mask, and $d_l$ is the dimensionality of $Q$, $K$, and $V$. We denote $\mathbf{A^i}$ as the attention matrix after $i$-th layer of LVLM, In this paper, we denote vision token attention $\mathbf{A}_{\text{img}} \in \mathbb{R}^{B \times H \times n}$ as the slice of the attention weights corresponding to the query from the last input token (the token immediately preceding the generated output) and the keys of all vision tokens $v$.

## 3.2 SPATIAL PERCEPTION BIAS

When given a blank white image and the open-ended prompt *"Please describe this image in detail"*, LVLMs are expected to distribute attention uniformly across the entire image. However, as shown in Figure 1a, the self-attention module assigns varying levels of attention to different spatial regions. For instance, LLaVA-1.5 places greater attention on later visual tokens, particularly near the bottom-right. This systematic attention bias reflects position-dependent sensitivity to visual features. We define this phenomenon as Spatial Perception Bias(SPB)—a systematic error in the self-attention module that skews attention weights toward specific spatial regions, leading to perception inconsistencies.

Xing et al. (2024) were the first to identify a similar issue, attributing it to the long-term decay effect of position encoding. Specifically, LVLMs tend to assign lower attention to tokens corresponding to the top-left region of an image compared to those in the bottom-right region. This asymmetric attention makes LVLMs more susceptible to object hallucination in the top-left region, where visual grounding is weaker. To mitigate this, they proposed reordering the visual token sequence to achieve a more balanced attention distribution. However, when comparing Figure 1(a–c), we find that SPB varies significantly across models and can result in unexpectedly high attention to arbitrary locations. Consequently, a predefined token reordering strategy cannot generalize well to LVLMs beyond LLaVA-1.5.

## 4 METHOD

### 4.1 UNIFORM ATTENTION CALIBRATION

To understand the core issue of spatial position bias, we can consider a simplified scenario. We hypothesize that an ideal model, when presented with a meaningless image (e.g., a blank white image), should distribute its attention uniformly across all visual tokens. Any deviation from this uniformity can be interpreted as a form of inherent model bias.

This leads to a straightforward calibration strategy we term Uniform Attention Calibration (UAC). The core idea is to first measure the model's vision token attention, $\tilde{\mathbf{A}}_{\text{img}}$, on a meaningless input (we use a blank white image by default). From this, we compute a static calibration matrix, $\mathbf{W}$, designed to counteract the observed bias:

$$\mathbf{W} = \frac{\text{avg}(\tilde{\mathbf{A}}_{\text{img}})}{\tilde{\mathbf{A}}_{\text{img}}} \tag{3}$$

where $\text{avg}(\cdot)$ denotes the average value over all elements of the matrix. During inference, this pre-computed matrix is applied as an affine transformation to the attention map of any given input image, $\mathbf{A}_{\text{img}}$, via an element-wise product:

$$\mathbf{A}'_{\text{img}} = \mathbf{W} \circ \mathbf{A}_{\text{img}} \tag{4}$$

By default, UAC is applied to a single self-attention layer in the decoder.

Despite its simplicity, this approach serves as a valuable proof of concept. As shown in Table 1, we observe that attention calibration effectively alleviates hallucination by mitigating SPB, particularly in the adversarial setting. This result supports our hypothesis that attention calibration is a promising direction. More results are provided in the Appendix.

However, the fundamental limitation of UAC remains its static, "one-size-fits-all" nature. Relying on a single bias profile is unlikely to work for diverse, content-rich inputs. Furthermore, such brute-force adjustments to the attention mechanism risk degrading the LVLM's general performance on other tasks. This motivates the need for a more robust and adaptive solutions.

| Method | Rnd↑ | Pop↑ | Adv↑ |
|--------|------|------|------|
| Baseline | 89.4 | 86.8 | 81.7 |
| VCD | 87.8 | 85.2 | 80.4 |
| OPERA | 90.0 | 86.9 | 81.8 |
| SID | 89.1 | 85.9 | 81.5 |
| CCA | 89.1 | 86.0 | 83.8 |
| UAC | **90.2** | **88.9** | **84.4** |

Table 1: POPE F1 scores on MSCOCO for LLaVA-1.5. "Rnd", "Pop" and "Adv" denote Random, Popular and Adversarial settings.

## 4.2 DYNAMIC ATTENTION CALIBRATION

To this end, we introduce Dynamic Attention Calibration (DAC). Instead of relying on a static calibration, DAC is a trainable, plug-and-play module designed to learn input-specific attention adjustments. It moves beyond a predefined rule by utilizing a contrastive learning framework Wu et al. (2018); Chen et al. (2020) to ensure the model produces consistent outputs regardless of an object's spatial position, thereby learning to mitigate SPB in a more effective and generalizable manner.

**DAC Design**     Motivated by the superior calibration performance of affine transformation in the field of uncertainty calibration Platt (1999), we introduce a lightweight trainable transformation $f$ to calibrate unreliable vision token attention weights before SoftMax function as $\mathbf{A}'_{\text{img}} = f(\mathbf{A}_{\text{img}})$, where $\mathbf{A}'_{\text{img}}$ denotes the calibrated vision token attention weights. Specifically, the transformation $f$ operates within the self-attention mechanism of the transformer decoder layers and consists of a small stack of linear transformations with ReLU activations. The details about building blocks can be found in the Appendix. The forward pass of DAC module can be defined as

$$\begin{aligned}
\mathbf{A}'_{\text{img}} &= f(\mathbf{A}_{\text{img}}) = \mathbf{g}_{L-1}\mathbf{W}_L + \mathbf{b}_L, \\
\mathbf{g}_i &= \text{ReLU}(\mathbf{g}_{i-1}\mathbf{W}_i + \mathbf{b}_i), \text{ for } i \in \{1, \dots, N-1\},
\end{aligned} \tag{5}$$

where $L$ denotes the layer number in DAC module, $\mathbf{W}_i \in \mathbb{R}^{D_i \times D_i}$ denotes the weight matrix of layer $i$, $\mathbf{b}_i \in \mathbb{R}^{D_i}$ denotes the bias vector, $\mathbf{g}_i$ represents the output of the $i$-th layer, and $\mathbf{g}_0 = \mathbf{A}_{\text{img}}$. The DAC module can be applied to any layer of the language model decoder, targeting the layers responsible for vision tokens processing.

**DAC Optimization**     With the DAC module in Eq. 5, a much stronger constraint can be imposed on vison token attention weights of LVLMs to alleviate the bias. Instead of the uniform constraint in UAC, we further propose to force the consistent outputs wherever the object locates in the image. The key idea is to ensure that the model maintains the same capability of identifying an object regardless of its position within the image. However, to impose such a constraint, it could be challenging to obtain sufficient training data variants with different object positions. Thus, we introduce a simple yet effective data augmentation technique inspired by the concept of instant discrimination Wu et al. (2018); Chen et al. (2020).

Formally, we randomly select 100 images from MSCOCO as our validation set, denoted as $\mathcal{D}_{\text{val}}$. Each image $V \in \mathcal{D}_{\text{val}}$ is paired with ground-truth annotations and their corresponding bounding boxes. The validation set $\mathcal{D}_{\text{val}}$ undergoes an augmentation process to produce the augmented calibration dataset $\mathcal{D}_{\text{cal}}$. Specifically, we crop the ground truth objects from the images using the annotations and bounding boxes provided, then apply random resizing and paste the cropped objects onto a pure white background as $V_{\text{crop}}$. For each $V_{\text{crop}}$, we generate balanced positive and negative query-label pairs, ensuring a well-balanced dataset. Additionally, we include annotations for the cropped images $V_{\text{crop}}$ to be utilized in instance discrimination tasks, as discussed later in the paper. The detailed augmentation process is summarized in the Appendix.

With sufficient augmented data from $\mathcal{D}_{\text{cal}}$, we propose leveraging contrastive learning to encourage LVLMs to focus on objects themselves rather than their absolute positions in the image. This approach ensures consistent outputs regardless of object position. By reducing reliance on positional cues, the model learns to robustly identify objects despite spatial transformations. Specifically, contrastive learning is formulated to increase the similarity between embeddings of the same object at different spatial locations while pushing apart the embeddings of different objects. We begin with an $\mathcal{D}_{\text{cal}}$ dataset and randomly sample a minibatch of $B$ examples. Each example then undergoes an additional augmentation process, resulting in a total of $2B$ augmented data points. Following the approach of

---

**Algorithm 1** DAC's Main Learning Algorithm

---

**Input:** Batch size $B$, constant $\tau$, frozen backbone networks $f(\cdot)$ and projection head $g(\cdot)$, augmentation distribution $\mathcal{T}$, augmented set $\mathcal{D}_{\mathrm{aug}} = \{(T_{\mathrm{aug}}, V_{\mathrm{crop}}, Y_{\mathrm{aug}})\}$
**for** sampled minibatch $\{(t_k, v_k, y_k)\}_{k=1}^{B} \in \mathcal{D}_{\mathrm{aug}}$ **do**
    **for** all $k \in \{1, \ldots, B\}$ **do**
        Draw one augmentation function $t \sim \mathcal{T}$
        $\tilde{v}_{2k-1} = v_k$; $z_{2k-1} = f(\tilde{v}_{2k-1})$; $\tilde{y}_{2k-1} = g(z_{2k-1})$; $y_{2k-1} = y_k$
        $\tilde{v}_{2k} = t(v_k)$; $z_{2k} = f(\tilde{v}_{2k})$; $\tilde{y}_{2k} = g(z_{2k})$; $y_{2k} = y_k$
    **end for**
    **for** all $i \in \{1, \ldots, 2B\}$ and $j \in \{1, \ldots, 2B\}$ **do**
        $s_{i,j} = z_i^\top z_j / (\|z_i\|\|z_j\|)$            # Pairwise similarity
    **end for**
    Compute the losses using: $\mathcal{L} = \mathcal{L}_{\mathrm{CE}} + \lambda \cdot \mathcal{L}_{\mathrm{CL}}$
    Update DAC parameters to minimize $\mathcal{L}$
**end for**
**Return:** Fine-tuned DAC

---

Wu et al. (2018), for each positive pair, we consider the remaining $2(B-1)$ augmented examples within the minibatch as negative examples. Given the embeddings $z_i$ and $z_j$ of the positive augmented pair $\tilde{v}_i$ and $\tilde{v}_j$, the contrastive loss can be expressed as:

$$\ell_{\mathrm{CL}}(i,j) = -\log \frac{\exp\left(\mathrm{sim}(\mathbf{z}_i, \mathbf{z}_j)/\tau\right)}{\sum_{k=1}^{2B} \mathbf{1}[k \neq i] \exp\left(\mathrm{sim}(\mathbf{z}_i, \mathbf{z}_k)/\tau\right)}, \tag{6}$$

where $B$ denotes the number of examples in a minibatch, $\mathrm{sim}(\cdot, \cdot)$ represents the cosine similarity, $\mathbf{1}_{[k \neq i]}$ is an indicator function, and $\tau$ is the temperature parameter. Combined with a cross-entropy (CE) loss, the final loss function is formulated as

$$\mathcal{L} = \mathcal{L}_{\mathrm{CE}}(F(T_{\mathrm{crop}}, V_{\mathrm{crop}}), Y_{\mathrm{crop}}) + \lambda \mathcal{L}_{\mathrm{CL}}, \tag{7}$$

where $F$ represents the model, $T_{\mathrm{crop}}$ and $V_{\mathrm{crop}}$ are the query and cropped image, $Y_{\mathrm{crop}}$ is the corresponding label, and $\lambda$ is a hyperparameter balancing the two losses. We optimize our DAC using Eq. 7 alongside instruction tuning, while keeping all other components frozen. The overall training process is summarized in Algorithm 1.

## 5 EXPERIMENT

### 5.1 SETUP

**Models and Baselines** We implement three representative LVLMs for evaluation: LLaVA-1.5 Shang et al. (2024), mPLUG-Owl2 Ye et al. (2024), and LLaVA-NeXT Liu et al. (2024c) at the 7B scale. Our methods are compared against five methods. Baseline responses are generated using the original LVLMs, while other techniques such as Visual Contrastive Decoding (VCD) Leng et al. (2023), OPERA Huang et al. (2023), Self-Introspective Decoding (SID) Huo et al. (2024), and Concentric Causal Attention (CCA) Xing et al. (2024) are included for comparative analysis. We adopt the default settings for OPERA, VCD, and SID. For CCA, we directly use the provided weights. For each compared method, except OPERA, which uses beam search (beam size 5), we use greedy decoding for polling-based tasks (POPE and MME), and nucleus sampling ($p = 1$) for open-ended generation tasks (CHAIR and LLaVA-Bench).

**Experiment Settings** Unless otherwise specified, we integrate the DAC module into two consecutive layers of the language model decoder. For all tasks, we use a fixed validation set $D_{\mathrm{val}}$, composed of 100 randomly selected MSCOCO images disjoint from any test set. For each image, we select up to three ground truth objects; if an image contains fewer than three objects, all available objects are included. Using these ground truth objects, we generate 10 cropped images per object, resulting in a dataset of approximately 5.4K $(T, V, Y)$ pairs. By default, the contrastive loss strength $\lambda$ is set to 0.01. To configure the DAC layer, we define 2–4 candidate layer buckets and select the setting when

validation on $D_{val}$ is applicable; otherwise, we adopt the same setting as used in the POPE MSCOCO Random.

For LLaVA-1.5, we fine-tune the DAC module on the $D_{cal}$ dataset using a learning rate of $3 \times 10^{-6}$, a batch size of 8, and gradient accumulation steps of 4. The training takes approximately 40 minutes on two NVIDIA RTX 4090 GPUs. We apply attention calibration to the vision token attention $\mathbf{A}_{img}$, computed with the last input token as the query. Additional experimental details can be found in the Appendix.

## 5.2 EVALUATION RESULTS

**POPE**   Polling-based Object Probing Evaluation (POPE) Li et al. (2023b) is a method designed to assess object hallucination in LVLMs. It evaluates model performance by querying the presence of specific objects in images using yes-or-no questions. POPE employs three strategies for sampling negative objects: Random, Popular, and Adversarial (refer to Li et al. (2023b) for details). Our evaluation utilizes two datasets: MSCOCO Lin et al. (2014) and A-OKVQA Schwenk et al. (2022). For each evaluation setup, every subset includes 3,000 questions across 500 images, resulting in a total of 18,000 yes-or-no questions. The evaluation pivots on two key metrics: Accuracy and the F1 score. DAC achieves the highest accuracy and F1 scores across most datasets and sampling setups, as shown in Table 2. Specifically, DAC delivers an average improvement of 1.01% in accuracy and 0.74% in F1 score for Random sampling, 2.19% in accuracy and 1.49% in F1 score for Popular sampling, and 2.41% in accuracy and 1.13% in F1 score for Adversarial sampling, compared to the next best existing approach. Notably, DAC achieves the largest accuracy gain in the more challenging Adversarial setting by effectively suppressing spurious visual cues that are unrelated to the target object.

**CHAIR**   The Caption Hallucination Assessment with Image Relevance (CHAIR) metric Rohrbach et al. (2018) is specifically designed to assess object hallucinations in image captioning tasks. CHAIR quantifies the degree of hallucinations in a generated image caption by calculating the proportion of objects mentioned in the caption that are not present in the ground truth label pool. Two common variants of CHAIR are defined: $C_S$ and $C_I$, which measure hallucination at the instance and sentence levels, respectively. These metrics are formulated as follows:

$$C_S = \frac{|\text{hallucinated objects}|}{|\text{all mentioned objects}|}, \quad C_I = \frac{|\text{captions with hallucinated objects}|}{|\text{all captions}|}$$

Lower values of $C_S$ and $C_I$ indicate better performances. Following Huang et al. (2023); Huo et al. (2024), we randomly select 500 images from MSCOCO validation set and query LVLMs using the prompt: *"Please describe this image in detail."* To ensure a fair evaluation, we limit the maximum number of new tokens to 512 when generating descriptions. As shown in Table 3, our method demonstrates effective improvements. Notably, on $C_S$, DAC achieves a significant 38.14% improvement across models compared to the next best approach. The superior performance of our method on CHAIR metrics highlights its effectiveness in mitigating hallucinations in open-ended generation settings.

**MME**   The MME benchmark Fu et al. (2024) provides a comprehensive framework for evaluating LVLMs across multiple dimensions. It includes ten perception-related subtasks and four cognition-focused tasks. Following Leng et al. (2023); Yin et al. (2023), we evaluate four perception subtasks that assess object-level and attribute-level hallucinations, specifically measuring object existence, count, position, and color. Table 4 presents the performance of our method, DAC, on the MME hallucination subset using LLaVA-1.5. DAC achieves a notable improvement of 16.16% over the baseline and 2.34% over the current state-of-the-art hallucination mitigation approaches, demonstrating its effectiveness in enhancing the general perception capabilities of LVLMs.

**GPT4V-Aided Evaluation**   We evaluate our approach on LLaVA-Bench Liu et al. (2024d), a benchmark comprising 30 images paired with a total of 90 questions. LLaVA-Bench is designed to assess the ability of models to generate coherent and contextually accurate responses for vision-language tasks. It categorizes questions into three types: conversation, detailed description, and complex reasoning. Following prior works Liu et al. (2024d); Huang et al. (2023), we prompt these models to generate responses and use the text-only GPT-4 Achiam et al. (2023) as the judge to rate

| Dataset | | | MSCOCO | | A-OKVQA | |
|---|---|---|---|---|---|---|
| Model | Setting | Method | Accuracy↑ | F1 Score↑ | Accuracy↑ | F1 Score↑ |
| LLaVA-1.5 | Random | Baseline | 89.63 | 89.74 | 87.30 | 88.49 |
| | | VCD | 87.53 | 87.81 | 85.00 | 86.49 |
| | | OPERA | 89.87 | 89.95 | 87.27 | 88.50 |
| | | SID | 89.43 | 89.08 | 87.30 | 88.00 |
| | | CCA | 89.77 | 89.05 | **90.00** | 90.11 |
| | | DAC | **90.83** | **90.60** | 89.70 | **90.33** |
| | Popular | Baseline | 86.23 | 86.82 | 80.30 | 83.2 |
| | | VCD | 84.43 | 85.20 | 77.50 | 81.07 |
| | | OPERA | 86.30 | 86.88 | 80.47 | 83.38 |
| | | SID | 85.93 | 85.94 | 82.00 | 83.80 |
| | | CCA | 89.77 | 89.05 | **85.45** | 85.01 |
| | | DAC | **89.50** | **89.10** | 83.96 | **85.52** |
| | Adversarial | Baseline | 79.70 | 81.71 | 69.33 | 76.10 |
| | | VCD | 78.13 | 80.38 | 67.90 | 75.01 |
| | | OPERA | 79.77 | 81.77 | 69.20 | 76.09 |
| | | SID | 80.43 | 81.47 | 72.93 | 77.48 |
| | | CCA | 83.97 | 83.82 | 74.77 | 78.32 |
| | | DAC | **84.12** | **84.42** | **75.42** | **79.21** |
| mPLUG-Owl2 | Random | Baseline | 86.27 | 86.88 | 81.57 | 83.89 |
| | | VCD | 84.40 | 84.79 | 82.53 | 84.16 |
| | | OPERA | 86.23 | 86.84 | 81.53 | 83.86 |
| | | SID | 86.30 | 86.86 | 83.53 | 85.28 |
| | | DAC | **87.71** | **87.57** | **86.56** | **87.24** |
| | Popular | Baseline | 80.73 | 82.52 | 75.97 | 79.98 |
| | | VCD | 81.00 | 81.12 | 75.70 | 79.21 |
| | | OPERA | 80.70 | 82.48 | 75.93 | 79.94 |
| | | SID | 81.27 | 82.82 | 77.47 | 80.89 |
| | | DAC | **87.57** | **84.96** | **82.83** | **83.47** |
| | Adversarial | Baseline | 76.17 | 77.69 | 67.37 | 74.63 |
| | | VCD | 77.10 | 77.00 | 68.80 | 74.85 |
| | | OPERA | 76.87 | 78.01 | 67.30 | 74.58 |
| | | SID | 77.27 | 79.89 | 68.93 | 75.43 |
| | | DAC | **82.58** | **82.32** | **75.88** | **77.78** |
| LLaVA-NeXT | Random | Baseline | 91.27 | 90.76 | 91.80 | 92.07 |
| | | VCD | 91.30 | 90.80 | 91.80 | 92.07 |
| | | OPERA | 91.36 | 90.80 | 91.77 | 92.03 |
| | | SID | 91.20 | 90.73 | 91.73 | 92.01 |
| | | DAC | **91.63** | **91.32** | **92.37** | **92.47** |
| | Popular | Baseline | 88.60 | 88.27 | 87.17 | 88.13 |
| | | VCD | 88.63 | 88.31 | 87.20 | 88.15 |
| | | OPERA | 88.65 | 88.60 | 87.20 | 88.15 |
| | | SID | 88.60 | 88.30 | 86.87 | 87.88 |
| | | DAC | **89.27** | **89.14** | **89.13** | **89.62** |
| | Adversarial | Baseline | 85.50 | 85.54 | 77.47 | 80.87 |
| | | VCD | 85.53 | 85.58 | 77.53 | 80.90 |
| | | OPERA | 85.10 | 85.75 | 77.21 | 80.62 |
| | | SID | 85.87 | **85.89** | 77.33 | 80.77 |
| | | DAC | **86.00** | 85.71 | **78.80** | **81.56** |

Table 2: POPE results. All results use greedy decoding, except OPERA (beam search), and are either reported from prior work or re-implemented using official code. Best performance in each setting is shown in **bold**.

these responses. The results on LLaVA-1.5 are presented in Table 5. Our method demonstrates strong performance across all question type. These results highlight the effectiveness of our approach at preserving language understanding and generation capabilities while significantly mitigating object hallucination.

| Setting | LLaVA-1.5 | | LLaVA-NeXT | |
|---|---|---|---|---|
| | $C_S\downarrow$ | $C_I\downarrow$ | $C_S\downarrow$ | $C_I\downarrow$ |
| Baseline | 51.3 | 16.8 | 42.6 | 14.1 |
| VCD | 48.0 | 14.3 | 41.3 | 12.9 |
| OPERA | 45.2 | 12.7 | 39.4 | 11.8 |
| SID | 45.0 | **11.7** | 38.4 | 11.4 |
| CCA | 48.6 | 13.4 | – | – |
| DAC | **30.6** | 12.3 | **21.4** | **10.2** |

Table 3: CHAIR on 500 MSCOCO images (max seq len 512). All results use nucleus sampling ($p$=1), except OPERA (beam search).

| Setting | Object-level | | Attribute-level | | Total↑ |
|---|---|---|---|---|---|
| | *exist.*↑ | *count*↑ | *pos.*↑ | *color*↑ | |
| Baseline | 175.67 | 124.67 | 114.00 | 151.00 | 565.33 |
| VCD | 184.66 | 138.33 | 128.67 | 153.00 | 604.66 |
| OPERA | 180.67 | 133.33 | 123.33 | 155.00 | 592.33 |
| SID | 190.00 | 148.33 | 128.33 | **175.00** | 641.66 |
| CCA | 190.00 | 148.33 | 128.33 | **175.00** | 641.66 |
| DAC | **195.00** | **158.33** | **133.33** | 170.00 | **656.67** |

Table 4: MME hallucination subset (greedy decoding; OPERA uses beam search).

| Method | Complex↑ | Details↑ | Conv↑ | Average↑ |
|---|---|---|---|---|
| Baseline | 66.3 | 46.7 | 68.7 | 60.6 |
| VCD | 69.6 | 51.6 | 57.3 | 61.6 |
| OPERA | 66.4 | **56.9** | 44.0 | 61.3 |
| SID | 66.7 | 51.3 | 66.3 | 60.4 |
| CCA | 66.1 | 53.9 | 69.4 | **64.3** |
| DAC | **70.3** | 50.0 | **72.7** | **64.3** |

Table 5: LLaVA-Bench results. The results are re-implemented using the official code and evaluated with the latest available text-only GPT-4 API. Scores are normalized by the total possible score. The best performances within each setting are highlighted in **bold**.

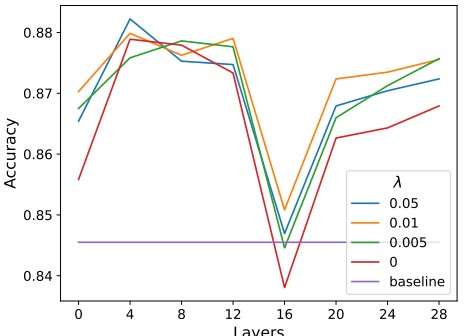

Figure 2: Performance of DAC under different settings of $\lambda$ and $N_{\text{DAC}}$.

### 5.3 ABLATION STUDY

**Hyperparameter** We analyzed two key hyperparameters: the contrastive loss strength $\lambda$ and the decoder layers $N_{DAC}$ to which DAC is applied. As shown in Figure 2, DAC consistently outperforms the baseline across most settings.

The contrastive learning component is critical for achieving performance gains. Our ablation study clearly demonstrates this: when the component is removed entirely by setting $\lambda = 0$, the model is fine-tuned only on the CE loss and yields the lowest performance among all tested settings. While excessively high values can degrade generative capabilities, performance is stable across a range of settings near the optimum. Our experiments indicate that $\lambda = 0.01$ achieves the best performance, with negligible differences for nearby values. For consistency, we adopt $\lambda = 0.01$ for all experiments.

Our method offers flexibility in choosing the decoder layers for applying the contrastive loss. Our results show that there is a wide range of effective choices. In practice, we follow a standard procedure: we identify 2–4 candidate pairs of consecutive decoder layers (e.g., layers 4–5 or 20–21) as brackets and select the best setting based on validation performance on $D_{\text{val}}$, if applicable.

## 6 CONCLUSION AND LIMITATION

This paper investigates object hallucination in LVLMs and identifies SPB as a key contributor, characterized by an imbalance in vision token attention that causes unequal focus across spatial regions and varies across models. This bias distorts object perception, amplifies sensitivity to misleading visual cues, and increases the risk of hallucination, compromising reliability in real-world settings. A straightforward UAC experiment confirms that mitigating SPB effectively reduces hallucination. Building on this, we introduce DAC, a learnable module that dynamically refines attention weights within the self-attention mechanism. Extensive evaluation confirms that DAC reduces hallucinations and enhances perception, highlighting attention calibration as a promising mitigation strategy.

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

## A  THE USE OF LARGE LANGUAGE MODELS

In preparing this manuscript, we used a Large Language Model (LLM) as a writing tool to enhance the clarity and presentation of the text. Its role was focused on linguistic improvements. In particular, the LLM was applied to:

- Refine sentences and paragraphs for improved readability and conciseness.
- Correct grammar, spelling, and punctuation.
- Strengthen the logical flow and transitions across sentences.

## B  ADDITIONAL UAC RESULTS

| Setting | POPE MSCOCO | | | CHAIR | | MME↑ |
|---------|------|------|------|-------------|-------------|-------|
| | $Rnd$↑ | $Pop$↑ | $Adv$↑ | $C_S$↓ | $C_i$↓ | |
| Baseline | 89.7 | 86.8 | 81.7 | 51.3 | 16.8 | 565.3 |
| VCD | 87.8 | 85.2 | 80.4 | 48.0 | 14.3 | 604.7 |
| OPERA | 90.0 | 86.9 | 81.8 | 45.2 | 12.7 | 592.3 |
| SID | 89.1 | 85.9 | 81.5 | 45.0 | **11.7** | 641.7 |
| CCA | 89.1 | 86.0 | 83.8 | 48.6 | 13.4 | 641.7 |
| UAC | 90.2 | 87.6 | 83.7 | 49.0 | 14.9 | 638.3 |
| DAC | **90.6** | **89.1** | **84.4** | **30.8** | 12.7 | **656.7** |

Table 6: Results on POPE MSCOCO, CHAIR, and MME hallucination subsets. "Rnd" "Pop" and "Adv" represent the Random, Popular, and Adversarial settings, respectively. On POPE MSCOCO, results are reported as F1 scores. The best performances within each settings are highlighted in **bold**.

To address the SPB inherent in LVLMs, we propose a toy example method Uniform Attention Calibration (UAC). UAC recalibrates biased attention by estimating SPB from a meaningless input. We evaluate this method using LLaVA-1.5 on the POPE MSCOCO, CHAIR, and MME benchmarks, following the same experimental setup as in our other comparisons. As summarized in tab:pope-wrap, UAC achieves the best overall performance on POPE MSCOCO compared to current state-of-the-art methods, surpassing other training-free approaches by a substantial margin. On the MME dataset, UAC attains competitive results. However, on the open-ended generation benchmark CHAIR, UAC falls short of the top performers. We attribute this to its reliance on a single meaningless image bias for calibration, which, while effective for structured tasks, may degrade generation quality in open-ended settings by limiting the model's ability to adapt to diverse contextual variations.

## C  DAC ARCHITECTURE

Detailed Dynamic Attention Calibration(DAC) applied to each layer of vision token attention is shown in Figure 3.

## D  DETAILED EXPERIMENTAL SETTINGS

Following the setup described in the main paper, we fix the contrastive–loss weight at $\lambda = 0.01$. The learning rates are set to $3 \times 10^{-6}$ for LLaVA–1.5, $4 \times 10^{-5}$ for MPLUG–Owl2, and $8 \times 10^{-7}$ for LLaVA–Next. Implementation details for $N_{\text{DAC}}$ on POPE are provided in Table 7, while those for CHAIR, MME, and LLaVA-Bench are listed in Table 8.

The application of DAC varies across models. For LLaVA-1.5 and LLaVA-NeXT, DAC is applied to the last token before prediction. For mPLUG-Owl2, DAC is applied to all tokens except system tokens, i.e., after the image starting position. For LLaVA-1.5 and LLaVA-NeXT, DAC consists of two layers with a hidden dimension of 576, which matches both the input and output dimensions. For mPLUG-Owl2, DAC is set to three layers with a hidden dimension of 576 to maintain a similar capacity.

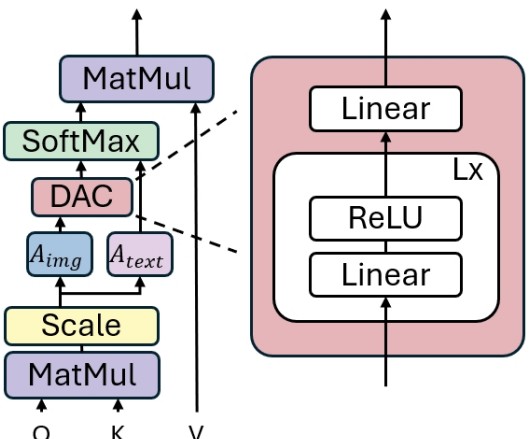

Figure 3: The Dynamic Attention Calibration (DAC) architecture consists of a small stack of linear transformations with ReLU activation, operating within the self-attention mechanism of transformer decoder layers to calibrate vision tokens attention.

| Model | Parameters | POPE | | |
| | | Rnd | Pop | Adv |
|---|---|---|---|---|
| LLaVA-1.5 | MSCOCO | 20, 21 | 4, 5 | 4, 5 |
| | AOKVQA | 20, 21 | 4, 5 | 4, 5 |
| MPLUG-Owl2 | MSCOCO | 12, 13 | 12, 13 | 12, 13 |
| | AOKVQA | 12, 13 | 12, 13 | 12, 13 |
| LLaVA-NeXT | MSCOCO | 16, 17 | 16, 17 | 16, 17 |
| | AOKVQA | 28, 29 | 28, 29 | 28, 29 |

Table 7: Optimal settings of DAC applied layers $N_{DAC}$ on POPE evaluation. "Rnd", "Pop" and "Adv" represent the Random, Popular, and Adversarial settings, respectively.

## E    DIFFERENT SAMPLING STRATEGIES

Table 9 presents an ablation study on various sampling strategies conducted on the POPE-Random dataset using LLaVA-1.5. In addition to the greedy decoding baseline discussed in the main paper, the study evaluates five alternative strategies: Top-P sampling (p = 0.9 and p = 1), Top-K sampling (k = 50), Top-K sampling with temperature scaling (k = 50, temperature = 0.7), and direct sampling (temperature = 1). The results show that applying DAC consistently reduces hallucination and enhances overall model performance across all decoding methods, underscoring the robustness and generalizability of DAC in mitigating hallucinations under diverse sampling conditions.

The augmentation process consists of the following steps:

- For each annotated object in $V$:
    - Crop the region defined by its bounding box.
    - Randomly resize the cropped object to a minimum size of $(H/14) \times (W/14)$ pixels (the typical size of an image patch) and a maximum size of $(H/2) \times (W/2)$, where $H$ and $W$ are the height and width of the original image $V$.
    - Replace the background of the cropped object with pure white, resulting in $V_{crop}$
- For each cropped object $V_{crop}$:
    - Generate a corresponding positive query $T_{pos}$ that describes the cropped object and assign the label $Y_{pos} = $ yes. Obtaining positive query-label pair: $(T_{pos}, V_{crop}, Y_{pos})$
    - Generate a ground-truth negative query $T_{neg}$, which refers to an object not present in the image, and assign the label $Y_{neg} = $ no. Obtaining negative query-label pair: $(T_{neg}, V_{crop}, Y_{neg})$

| Model | CHAIR | MME | LLaVA-Bench |
|-------|-------|-----|-------------|
| LLaVA-1.5 | 5, 6 | 20, 21 | 20, 21 |

Table 8: Optimal settings of DAC applied layers $N_{DAC}$ on CHAIR, MME, and LLaVA-Bench using LLaVA-1.5.

| Setting | Baseline | | VCD | | DAC | |
|---------|----------|------|------|------|------|------|
| | Acc↑ | F1↑ | Acc↑ | F1↑ | Acc↑ | F1↑ |
| Top-$p$ = 0.9 | 84.91 | 83.05 | 87.82 | 87.31 | **88.60** | **88.18** |
| Top-$p$ = 1.0 | 84.77 | 82.28 | 86.84 | 86.83 | **87.77** | **87.50** |
| Top-$k$ = 50 | 83.04 | 81.05 | 87.49 | 86.92 | **87.57** | **87.19** |
| Top-$k$, $t$=0.7 | 85.17 | 83.38 | 85.13 | 85.94 | **89.47** | **89.23** |
| Sample, $t$=1 | 83.29 | 81.33 | 87.73 | 87.16 | **88.17** | **87.86** |

Table 9: Various sampling strategies conducted on the POPE-Random dataset using LLaVA-1.5.

  – Each cropped image $V_{\text{crop}}$ results in one positive query-label pair and one negative query-label pair, ensuring a balanced augmented set.

Let $I$ represent the number of original images in the calibration set $\mathcal{D}_{\text{cal}}$, $J$ represent the average number of annotated ground-truth objects per image $V$, and $K$ represent the number of crops generated per object. The total size of the augmented dataset is: Total size of $\mathcal{D}_{\text{aug}} = I \cdot J \cdot K \cdot 2$

# F   SPB ON OTHER BLANK IMAGES

Additional case studies of SPB under different vision and prompt inputs using LLaVA-1.5 are presented in Figures 4–10.

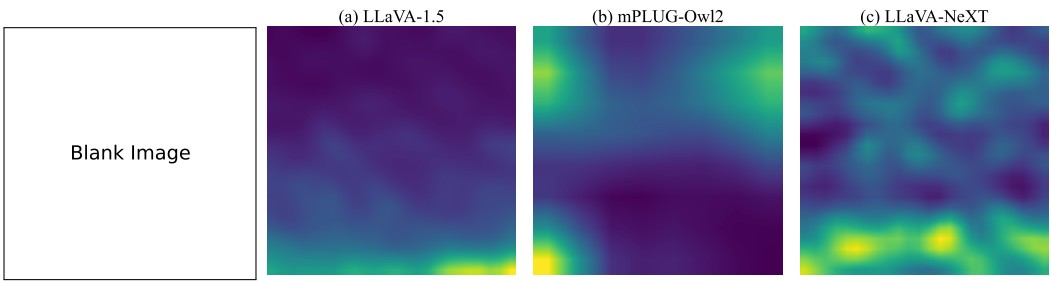

Figure 4: Vision tokens attention weights during the decoding process for different models on a blank white image in response to the polling prompt: "Is there a bear in the image?"

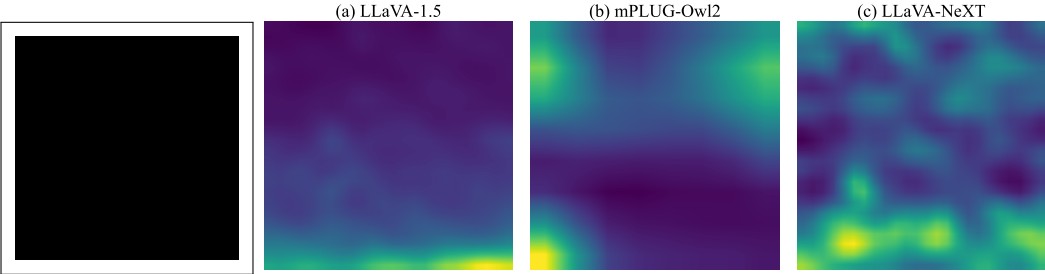

Figure 5: Vision tokens attention weights during the decoding process for different models on a blank black image in response to the polling prompt: "Is there a bear in the image?"

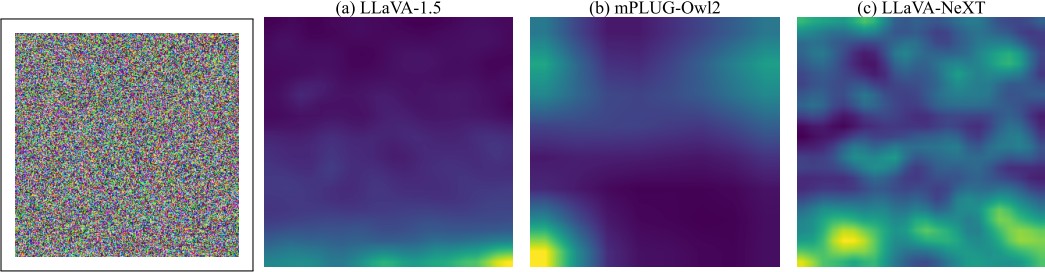

Figure 6: Vision tokens attention weights during the decoding process for different models on a blank noise image in response to the polling prompt: "Is there a bear in the image?"

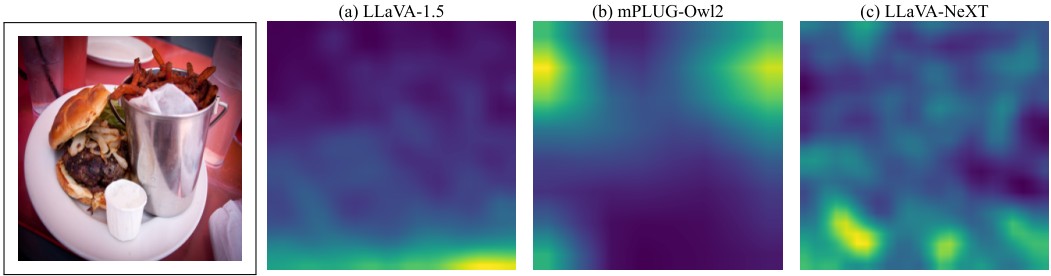

Figure 7: Vision tokens attention weights during the decoding process for different models on an actual image in response to the polling prompt: "Is there a bear in the image?"

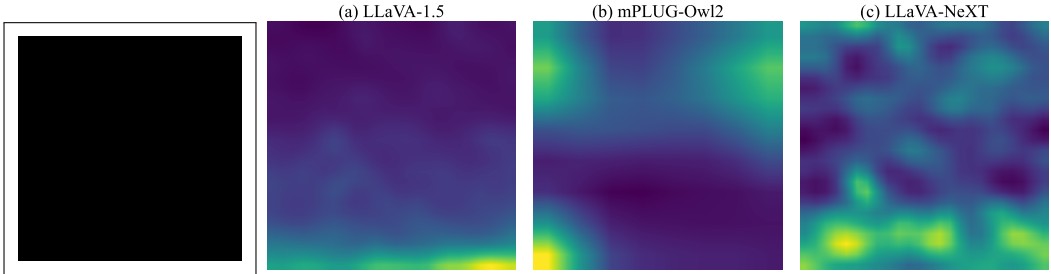

Figure 8: Vision tokens attention weights during the decoding process for different models on a blank black image in response to the open-ended prompt: "Please describe this image in detail."

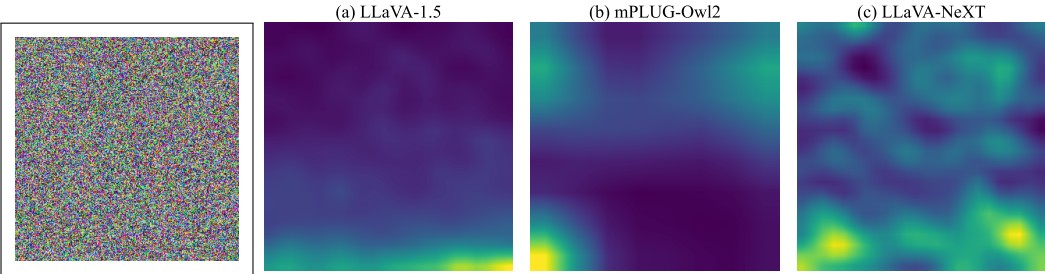

Figure 9: Vision tokens attention weights during the decoding process for different models on a blank noise image in response to the open-ended prompt: "Please describe this image in detail."

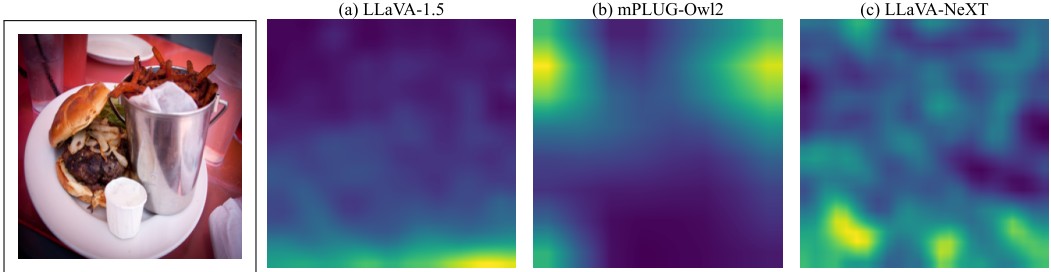

Figure 10: Vision tokens attention weights during the decoding process for different models on an actual image in response to the open-ended prompt: "Please describe this image in detail."

