# OpenReview forum: "Attention Calibration for Reducing Hallucination in Large Vision-Language Models"
_ICLR.cc/2026/Conference — ICLR 2026 Conference Withdrawn Submission_

### Official Review · Reviewer_scRb · 2025-10-29

**Soundness:** 1
**Presentation:** 3
**Contribution:** 1
**Rating:** 2
**Confidence:** 5

**Summary:**

This paper argues that hallucinations in LVLMs arise from positional bias in visual tokens. To address this issue, the authors introduce a calibration technique designed to compensate for such bias. The method is implemented on three LVLMs: LLaVA-1.5, mPLUG-Owl2, and LLaVA-NeXT, and evaluated across four hallucination benchmarks: POPE, MME, CHAIR, and LLaVA-Bench.

**Strengths:**

- This work focuses on two key issues in LVLMs: 1) hallucination 2) positional bias in visual attention.
- Addressing hallucination through intervention at inference is an efficient choice, compared to techniques that require heavy training.
- The paper is easy to follow.

**Weaknesses:**

* The opening statement in the abstract seems inaccurate. Current LVLMs are not particularly strong at reasoning — in fact, limited reasoning ability is one of their fundamental weaknesses, independent of their susceptibility to hallucination. (l.011: "...(LVLMs) exhibit impressive multimodal reasoning capabilities...")

* In Figure 1, the attention maps (a–c) are derived from a blank image, which contains no meaningful visual information. Consequently, these maps offer limited insight. It would be more informative to visualize attention patterns using real images with meaningful objects — do the attentions remain concentrated in the same regions, or do they shift elsewhere?

* The baseline models used (LLaVA 1.5 and mPLUG-Owl2) are relatively outdated. It would strengthen the paper to evaluate the proposed method on more recent LVLMs such as Qwen-VL 2.5, InternVL 2.5, or BLIP-3.

* The benchmarks employed in this study are not sufficiently robust. For instance, POPE assesses object existence on only 500 images and does not account for other forms of hallucination, such as object attributes or relations. Similarly, CHAIR evaluates just 500 MSCOCO images with limited ground-truth annotations. LLaVA-Bench includes only 30 images and relies on GPT-4 as a judge, which is also unreliable. Overall, these benchmarks provide a weak basis for evaluation.

* Several results appear to be missing. Specifically, there are no results on LLaVA-Bench and MME for LLaVA-Next.

* "Specifically, LVLMs tend to assign lower attention to tokens corresponding to the top-left region of an image compared to those in the bottom-right region. This asymmetric attention makes LVLMs more susceptible to object hallucination in the top-left region, where visual
grounding is weaker." -- this statement is not backed by any significant proof. Even the examples shared in Figure 1 do not satisfy this argument.

* There are issues with citation format.

**Questions:**

- Please see weaknesses.

---

### Official Review · Reviewer_mYBw · 2025-10-29

**Soundness:** 3
**Presentation:** 2
**Contribution:** 2
**Rating:** 4
**Confidence:** 4

**Summary:**

This paper addresses the persistent problem of object hallucination in LVLMs—cases where a model generates visual descriptions that are misaligned with the actual image content. The authors identify Spatial Perception Bias (SPB), a form of systematic positional bias in the visual attention of LVLMs, as a core cause of hallucination. To investigate this bias, they first conduct a simple test by inputting blank images into different LVLMs, revealing significant position-dependent variations in attention distribution. Building on this finding, the paper first introduces a static correction method, Uniform Attention Calibration (UAC), which removes position-based bias by adjusting attention maps using data from a blank image. Extending this idea, the authors propose Dynamic Attention Calibration (DAC), a lightweight, learnable, and plug-and-play module that employs contrastive learning to enforce positional invariance in visual attention dynamically. DAC is integrated directly into the self-attention layers of LVLM decoders and fine-tuned using paired and augmented image samples. Experiments across several benchmarks show that DAC significantly reduces hallucinations while improving multi-modal consistency and perceptual accuracy.

**Strengths:**

1. The author explores the factors contributing to hallucinations in LVLMs from the perspective of Spatial Perception Bias.

2. The Dynamic Attention Calibration (DAC) mechanism employs contrastive learning to enhance positional robustness, demonstrating significant effectiveness in practical applications.

3. DAC exhibits low computational cost, revealing its potential value as a general-purpose hallucination mitigation module.

**Weaknesses:**

1. The fundamental cause of spatial perception bias requires further analysis.
2. The method is relatively simple and lacks novelty. The contrastive learning approach relies excessively on augmented data, which may fail to adequately capture the complexity of visual scenes and spatial relationships.
3. There is a lack of visualization analysis, and no attention distribution is presented on real-world data.

**Questions:**

1. Is spatial perception bias primarily derived from the training set?
2. What is the difference between SPB and attention? I feel that conceptually they are the same.
3. If spatial perception and attention are conceptually consistent, then what is the theoretical difference compared with previous papers that addressed hallucination by altering the attention distribution?

---

### Official Review · Reviewer_5D7A · 2025-10-29

**Soundness:** 2
**Presentation:** 3
**Contribution:** 2
**Rating:** 4
**Confidence:** 5

**Summary:**

This paper proposes a dynamic attention calibration approach that uses contrastive learning to adjust vision token attention, aiming to mitigate spatial perception bias in LVLMs. Experimental results show that it helps reduce object hallucination to some extent.

**Strengths:**

1. The paper tackles an interesting and underexplored problem, positional perception bias, with clear motivation and visual evidence in figure 1.
2. The method builds on CCA and is clearly described and easy to follow.

**Weaknesses:**

1. The performance gain on POPE in Table 2 appears marginal, especially for LLaVA-Next (all within 1% gain). It’s unclear whether the improvement is beyond the standard deviation.

2. I’m concerned about the generalizability of the method since the calibration is trained and evaluated entirely on MSCOCO images. The paper would be stronger if it:
(1) Included POPE results on the GQA dataset, following the same setup as the original POPE paper; and
(2) Reported CHAIR results for the mPLUG-Owl2 model, which is already included in the POPE evaluation but not shown (even in the appendix).

3. Missing important baselines on attention calibration e.g. [1] from ACL 2025.

[1] Don’t Miss the Forest for the Trees: Attentional Vision Calibration for Large Vision Language Models

**Questions:**

1. Is there any analysis or qualitative result that specifically investigates why the positional perception bias emerges? I find this aspect less well studied and lacking deeper explanation. I agree with the previous CCA paper’s point that long-term decay in RoPE poses challenges for modeling cross-modal interactions over long spatial distances, but this work could be more insightful and stronger if it further analyzed the underlying causes of the positional perception bias.

2. The paper encourages LVLMs to focus more on the objects themselves rather than their absolute positions, however, positional information is still valuable for spatial reasoning. Is there any downside or potential loss of positional cues caused by the proposed calibration? A discussion on this tradeoff, along with extended results on the MME position task across models and baselines, would make the paper stronger

---

### Official Review · Reviewer_4JEV · 2025-11-02

**Soundness:** 3
**Presentation:** 1
**Contribution:** 2
**Rating:** 2
**Confidence:** 4

**Summary:**

The paper identifies spatial position bias as an imbalance in how vision tokens are attended across spatial regions and considers it as a major cause of hallucination in LVLMs. The authors validate this phenomenon using Uniform Attention Calibration by removing attention bias measured from blank images and show hallucination reduction. Based on this, authors propose Dynamic Attention Calibration (DAC) and considers it as a lightweight plug-and-play module that learns to dynamically adjust attention weights through contrastive learning. Experiments cover LLaVA-1.5, mPLUG-Owl2 and LLaVA-NeXT on evaluations including POPE, CHAIR, MME and LLaVA-Bench.

**Strengths:**

1. The paper addresses an important issue of object hallucination in LVLMs.
2. The proposed method DAC is simple, lightweight, and model-agnostic, requiring minimal fine-tuning.
3. Empirical results demonstrate improvements across multiple benchmarks and LVLM architectures.

**Weaknesses:**

1. Contrastive methods to reduce visual hallucination have been extensively studied in previous works. Related works are not being sufficiently discussed but instead the authors spent a large amount of texts on introducing VLMs even before CLIP, which are not the closest related works. Just to name a few directly related publications: (1) HALC: Object Hallucination Reduction via Adaptive Focal-Contrast Decoding. (2) Contrastive Region Guidance: Improving Grounding in Vision-Language Models without Training. A more rigorous discussion on past publications is required to clarify how this method meaningfully differs from (and outperforms) existing hallucination reduction approaches.

2. The evaluation relies heavily on older models like LLaVA-1.5, with minimal inclusion of more recent and representative LVLMs such as the InternVL, Qwen-VL (especially Qwen2.5-VL), Gemma3, or Kimi-VL-A3B series. The effectiveness shown on llava and the older models may not transfer to these current model. Even within the experiments, several evaluations omit LLaVA-Next or mPLUG-Owl2 (e.g., Tables 3–5). Evaluation also focused on old metrics, where benchmark like HallusionBench [1] should be considered.

3. A substantial portion of the paper is dedicated to general background material not directly related to hallucination mitigation. For example, the subsection on the self-attention mechanism (lines 162–173) merely reintroduces textbook content and adds little to the main contribution. Why is it necessary to spend a subsection on introducing the very basic attention mechanism??

In addition, citation formatting and consistency are highly problematic throughout. Overall, the writing and presentation quality fall short of ICLR standards.

[1] HallusionBench: An Advanced Diagnostic Suite for Entangled Language Hallucination and Visual Illusion in Large Vision-Language Models

**Questions:**

See weaknesses.

---

### Note · Authors · 2025-11-12

I have read and agree with the venue's withdrawal policy on behalf of myself and my co-authors.